# Adults’ Perception of Children’s Vowel Production

**DOI:** 10.3390/children9111690

**Published:** 2022-11-03

**Authors:** Tae-Jin Yoon, Seunghee Ha

**Affiliations:** 1Department of English Language and Literature, Sungshin Women’s University, Seoul 02844, Korea; 2Division of Speech Pathology and Audiology, Audiology and Speech Pathology Research Institute, Hallym University, Chuncheon-si 24252, Korea

**Keywords:** children’s speech, production and perception, monosyllabic words, vowel space, gender, sound change

## Abstract

The study examined the link between Korean-speaking children’s vowel production and its perception by inexperienced adults and also observed whether ongoing vowel changes in mid-back vowels affect adults’ perceptions when the vowels are produced by children. This study analyzed vowels in monosyllabic words produced by 20 children, ranging from 2 to 6 years old, with a focus on gender distinction, and used them as perceptual stimuli for word perception by 20 inexperienced adult listeners. Acoustic analyses indicated that F0 was not a reliable cue for distinguishing gender, but the first two formants served as reliable cues for gender distinction. The results confirmed that the spacing of the two low formants is linguistically and para-linguistically important in identifying vowel types and gender. However, a pair of non-low back vowels caused difficulties in correct vowel identification. Proximal distance between the vowels could be interpreted to result in the highest mismatch between children’s production and adults’ perception of the two non-low back vowels in the Korean language. We attribute the source of the highest mismatch of the two non-low back vowels to the ongoing sound change observed in high and mid-back vowels in adult speech. The ongoing vowel change is also observed in the children’s vowel space, which may well be shaped after the caregivers whose non-low back vowels are close to each other.

## 1. Introduction

Previous studies have provided important findings about children’s speech production development [1,2,3,4,5]. The collective body of information from previous studies has indicated that essentially all aspects of children’s speech demonstrate changes toward more adultlike characteristics of their ambient language over time [1]. Nevertheless, currently few studies examine the link between children’s production and inexperienced adults’ perception of speech sounds, especially taking into consideration the ongoing sound change such as the raising of the mid-back vowel /o/ [6].

Research has found that even infants as young as 5 months old rapidly modify their vocalizations in response to audio-visual recordings of vowels produced by an adult on television [7]. Ten-month-old infants, when they are exposed to different languages produce babbling with some language-specific vowel characteristics when measured acoustically [8,9]. These findings suggest that “young children have some ability to process their own vocal output and can link sensory patterns that they have seen and heard with sensory-motor patterns that they are attempting to produce” [10].

Regarding linguistic attributes of speech sounds, F0 and the lowest two formants (i.e., F1 and F2) are considered the most important acoustic features for linguistic categories and structures, especially for sonorant sounds such as vowels. For example, F0 plays a role in conveying prosodic prominence and phrasing [11,12], and the lowest two formants are acoustic cues for vowel types [13,14]. Although F0, F1, and F2 are among the most important acoustic features in shaping linguistic categories and structures, researchers have reported that at least for children speaking American English [15,16], F0 and the two formants are developed differently based on gender.

As for F0, it seems that both boys and girls speaking American English exhibit a similar F0 range. Because children’s vocal folds are much shorter than those of adults, children’s pitch is much higher than adults’. The non-distinction of gender based on F0 might be due to the influence of caregivers’ modification of speech style toward infants. In many languages, caregivers increase their pitch and expand their vowel spaces when producing infant-directed speech (IDS) [17,18]. These acoustic modifications have been shown to attract infant attention. The F0 range typically found in IDS for female adults overlaps with infant speech and IDS, making gender distinction based on F0 difficult [10]. The F0 differences between gender reportedly begin to emerge around 7 years of age. It is worth testing whether Korean children of both genders show similar developmental trajectories as American children in the F0 range.

The first two lowest formant frequencies have provided essential cues when investigating vowel identification since the study of [19], whose formant hypothesis has been dominant ever since. This approach has been supported by many subsequent studies and is often mentioned in studies exploring the role of formants and their characteristics in vowel perception. Unlike the lack of F0′s role in gender distinction, the vowel space fails to reveal an overlap between IDS and infant speech with respect to formant patterns [10]. Thus, formant patterns are shaped based on the children’s vocal tract [13], and gender differences can result in different formant structures. As the length of the vocal tract of boys is longer than that of girls, the vowel space based on F1 and F2 will be smaller for boys than girls [15,20]. The study by [14] cast doubts on the role of formants in gender discrimination. Their study measured vowel formants (F1 and F2) and vowel space among 3- to 6-year-old Korean children residing on Jeju Island. Despite the expectation that the vowel space for girls would be larger than for boys, Ref. [14] reported that no differences of vowel space emerged due to gender (*p* > 0.05). Thus, it is worth examining whether formants play a role in distinguishing the speech characteristics of boys from those of girls in standard Korean.

Infant-directed speech may also help children form the phonological system of their native language. Regarding the vowel system in Korean, two pairs of vowels are worth noting, as one pair (i.e., [e] and [æ]) completes a merger [14,21,22], and the other (i.e., [o] and [u]) reportedly undergoes a merger [6]. Thus, in both adults’ and children’s productions the vowel [e] in the Korean word [ke] ’crab’ and the vowel [æ] in [kæ] ’dog’ are now merged into a single (lower) mid-front vowel [e]. As for the non-low back vowels, [o] and [u] in Seoul Korean are undergoing a merger in the F1/F2 space, especially for female speakers [23]. In studies conducted before 2000, the Korean short vowels [a, ʌ, o, u, ɯ, i] were clearly distinguished in the vowel space, except for [e]–[æ] [21]. For example, the vowels [o, u] produced by male and female speakers in their 20s, as measured in the study of [21], maintained a sufficient distance from the vowel space, and the two vowels had a clear difference from the F1 value. On the other hand, more recent phonetics studies have reported changes in formant values implying that the vowel [o] is ascending [6,23]. For example, Ref. [23] examined the difference in the formant values of monophthongal vowels uttered by men and women in their 20s and 30s and found that the vowels [o] and [u] were close in the acoustic space. They also found a gender difference: In the case of men, there was a statistically significant difference in the F1 values of the two vowels, but no statistically significant difference was identified in the F1 and F2 values of female speakers. This implies that the ongoing sound change of [o] raising is led by young female speakers [24]. Ref. [6] reported evidence of a chain-like vowel shift possibly being underway in Seoul Korean that has caused [o] to lower its F1 and rise to become more like [u], which in turn might be becoming more fronted (cf. [25]).

Given the previous research, we hypothesize that children’s vowel production may be similar to adults’ vowel space, resulting in the ongoing sound change of [o]-raising. We also hypothesize that the proximity of [o] and [u] in the children’s vowel space may result in greater misperception by adult listeners. In order to test these hypotheses, this paper conducts acoustic analyses of vowels produced within monosyllabic words by children 2 to 6 years old and perceived by 20 inexperienced adult listeners. In the acoustic analysis, we analyzed boys’ and girls’ voices separately because the acoustic features may be different based on gender [15].

We aim to answer the following research questions: First, what are the gender-specific acoustic characteristics of vowels produced by Korean children 2 to 6 years old? This question is based on previous research on American children, which showed that children’s gender leads to different formant patterns distinct from F0. Thus, we wanted to see whether the same pattern emerges in Korean children before further acoustic analyses. Second, how correctly would inexperienced listeners transcribe targeted vowels that children produce in monosyllabic words—that is, would inexperienced listeners respond according to the manifested acoustic vowel space when it comes to the recognition of vowels flanked in familiar monosyllabic words? The second question is based on the asymmetric research endeavors of developmental studies. Research on the acquisition of segments tends to be on consonants, not vowels (cf. [5] and references therein). This study fills the gap by focusing on the perception of children’s vowel production

## 2. Materials and Methods

### 2.1. Participants

Data for this study were drawn from a large normative study of speech development being conducted in Korea [26]. The Hallym University Institutional Review Board approved the normative study and the following perception study. Research flyers were posted at local daycare or preschool communities and on social network sites. Signed consent forms were obtained from children’s caregivers or adult participants.

All the children were native Korean speakers in monolingual environments and attended daycare centers or preschools at the time of data collection The recordings of data were collected from 20 children (10 boys and 10 girls), whose average age was 52.4 months (range = 26~81 months; SD = 17.9 months). All the children were reported to have no diagnosed developmental difficulties as well as no language, hearing, or cognitive concerns by parents or caregivers. The recordings were divided into 5 data sets according to children’s age from 2 to 6 years and each data set consisted of recordings from 4 children.

Twenty adults participated in the study as listeners. They were all undergraduate students (6 males and 14 females) in their early 20s and native Korean speakers. None of the listeners reported any history of speech or hearing problems. The listeners were inexperienced in that they were not familiar with children’s pronunciation patterns and had not taken any phonetic or phonology classes.

### 2.2. Data Collection

The Korean Articulation and Phonology Profile (K-APP) [26] was administered to the children in a quiet space in the presence of their caregivers (a parent or a daycare teacher) and audio-recorded using a portable digital voice recorder (Sony ICD-UX400F, Sony Corporation, China).

Eighteen consonant-vowel-consonant (CVC) monosyllables in K-APP were selected for target words to control the effect of different syllable structures and word lengths on children’s articulation proficiency. Seven vowels and 19 consonants in Korean phonology were used for the target monosyllabic words but are not strictly controlled in terms of the consonant types occupying the onsets and codas. The target words are familiar to children. Due to the complete merger between [e] and [æ], we did not consider collecting words with different front mid vowels. The central high vowel [ɯ] is not included on the list due to the lack of familiar monosyllabic words. The target words are listed in Table 1.

Before recording, the research assistant checked whether children knew the target words. All the target words were familiar to children except for two words, [hak] for ‘crane’ and [ɹiŋ] for ‘ring’. The research assistant presented the pictures and names of the two target words to all children before recording.

Children were asked to look at pictures of the target words, which were presented on a laptop computer screen using PowerPoint, and spontaneously produce them. When a child had difficulty spontaneously producing certain target words, the research assistant presented the target words and then asked his/her to produce them as spontaneously as possible.

To analyze the F0 and formants from children’s vowel production and use auditory stimuli for the perception study, we selected exclusive recording samples in which children correctly produced the vowel of the target words. Praat [27] was used for the feature extraction. The speech samples were annotated using the TextGrid function in Praat, and a custom-made script was written to extract F0, F1, and F2 at the middle point of the targeted vowels, as shown in Figure 1. The F0 range was set from 75 to 600 Hz, and the formant ceiling was set to 7500 Hz. The setting was changed from the default values to reflect the shorter length of the vocal tract and vocal folds than those of adults. The exact values in the setting were chosen empirically after visually examining the F0 and formant trajectories of every speech sound.

### 2.3. Data Analysis

A linear mixed effects model was used to model the acoustic data. The program R (R Development Core Team, 2016, Vienna, Austria) [28] and the package *lmer* [29] were used for the data analyses. Acoustic properties of interest—that is, F0 (in semitone), F1, and F2 (in Hertz)—were treated as response variables. Linear mixed effects models were fit to each measure of F0, F1, and F2, respectively. Gender was treated as a fixed variable while individual child (Subj as a variable name) and type of vowels were treated as random variables. For F0, the following formula was used with the *lmer* package in R:F0model = *lmer*(F0 Gender + (1*|*Subj) + (1*|*Vowels), data = data, REML = FALSE)

Formants were modelled using the same formula, but with the different independent acoustic features of F1 and F2.

### 2.4. Perception Study

The aforementioned 20 listeners were randomly assigned to one of 5 listener groups. Each group was asked to listen to a speech stimulus set containing monosyllabic words produced by 4 children, as shown in Table 2. The empty cells in Table 2 indicate where no children’s speech was assigned in the age group. In the tableshows the age and gender distribution of the children. Each age group consisted of 4 children.

All listeners were tested in front of a computer in a quiet room. The stimuli were presented using PowerPoint and played through headphones (Britz W800BT, Britz International Corporation, Seoul, Korea). The listeners were allowed to adjust the audio volume to a comfortable level. The listeners were instructed that they would hear monosyllabic CVC words produced by young children. They were allowed to listen to each stimulus up to 2 times and were instructed to write down the monosyllabic words they understood. Each listener in a group completed the listening task, identifying a total of 288 tokens.

## 3. Results

### 3.1. Children’s Vowel Production

#### 3.1.1. F0 (In Semitone)

Table 3 reports the fixed effects, showing the estimates of the coefficients for gender (male compared to female, which is in the intercept), the standard errors of these estimates, and the t value of the coefficient (estimate/standard error). The results indicate that, although male speakers produce higher F0 than their female counterparts (21.6 semitones for male speakers vs. 19.5 semitones for female speakers), gender is not differentiated based on F0 alone (*p* > 0.1). The finding that F0 alone is an insufficient cue for gender distinction agrees with previous studies on American children [15]. Thus, Korean adult listeners are likely to experience difficulty in correctly identifying Korean children’s gender if only an F0 cue is provided.

The interpretation can be aided visually with the box and whisker plot in Figure 2:

#### 3.1.2. Formant

When fitting formants using the same fixed effect and random effects, gender had a statistically significant effect on F1 and F2, respectively, as can be seen in Table 4 for F1 and Table 5 for F2. The F1 result suggests that, on average, the first formant (i.e., F1) of boys’ speech samples was 728.34 Hz, whereas that of girls’ speech samples was higher at 825.73 Hz. Likewise, the second formant (i.e., F2) of boys’ speech samples was 1486.3 Hz whereas that of girls’ samples was higher at 1645.57 Hz on average. These differences in gender in the first two formants were statistically significant (*p* < 0.05 for both F1 and F2). If we assume that boys’ vocal tract is a bit longer than girls’ vocal tract, it explains boys’ lower formant values. The statistical significance of gender differences in formants is in line with the reports that American children’s gender can be discerned based on formants [15]. Our results demonstrate that Korean children’s gender can also be distinguished based on formant-based acoustic features. 

The mean F1 and F2 values were plotted for each gender, as shown in Figure 3. As Figure 3 indicates, the vowel space of male speakers was smaller than that of female speakers. In general, the high front vowel was more posterior for male speakers than for female speakers while the low vowel was higher for female speakers than for male speakers. However, the high back vowel was higher (but not more posterior) for male speakers than for female speakers. Considering that these vowels are corner vowels, the male speakers occupied a smaller vowel space than female speakers.

An interesting observation was that the distance between [u] and [o] was the closest in the vowel space, regardless of when the vowels were collapsed for both genders (left in Figure 3) and for each gender (right in Figure 3). It may well be the case that the proximity between a pair of vowels affects the perception of the vowel pair.

### 3.2. Adult’s Perception of Children’s Vowel Production

We conducted a perception experiment on children’s production of monosyllabic target words with inexperienced adult speakers in their 20s. A detailed description of perception experiments and the results of consonantal perception were reported in [30]. To help explain the results of the current study, two significant findings from [30] are summarized. The first relates to the agreement rate between children’s production of the target words in Table 1 and inexperienced listeners’ perception of the production. The agreement rate among listener groups reveals that Group A had a lower agreement rate (42.7%) than all other groups (from 58.7% to 64.2%). However, a pair-wise comparison of the listener groups indicated no statistical significance except for Groups A and D (*p* < 0.05). As expected, the agreement rate depended on the age of children producing the target words and the analytic conditions, which included three conditions: when all target words are correctly perceived by listeners, when only the CV (i.e., onset and nucleus) in the CVC form is correctly identified, and when only onset segments are correctly identified. As age increases and the condition is relaxed, the agreement rate goes up. In addition, consonants seemed to be more prone to misperception than vowels. For example, words like /kim/ ‘seaweed’ tended to be mispronounced as [tɕim] (palatalization) or [ki] (coda deletion) with the vowel intact. Neither [kam] (vowel lowering) nor [kum] (vowel backing) for /kim/ ‘seaweed’ was attested.

We examined which vowel types were prone to misperception. We extracted cases in which onset and coda were correctly identified, but only the vowel was incorrectly identified. When we focused on the vowels only, only 20 cases between the target vowel and listeners’ perception were mismatched, which are graphically represented in Figure 4. As the figure indicates, most of the mismatched cases were counted as one, which may be attributed to an error. However, we can see that the chance of [o] (‘ㅗ’) being perceived as [u] (‘ㅜ’) was relatively high. The proximity between [o] and [u] in the vowel space (see Figure 3) provides an explanation for adults’ misperception of children’s production of [o] as [u]. Specifically, children’s vowel production is acquired under the influence of their caregivers, whose non-low back vowels are in proximal distance. Thus, the inexperienced adult listeners had difficulty correctly distinguishing the mid back vowel [o] from the high back vowel [u].

## 4. Discussion

A newborn baby’s larynx descends from its elevated position at birth over a period of four years. The length of the vocal cords to the lips at birth is approximately 6–8 cm, but grows to 15–18 cm in adulthood [16]. Consequently, children have a higher pitch and formant frequency than adult speakers because the length of the vocal cords and vocal tract is shorter in children than in adults. Similarly, the length from vocal cords to lips is 14–14.5 cm on average for adult women and 17–18 cm on average for adult men, meaning women have a higher formant frequency than men. The present study found that F0 was not significantly different between the genders, which is consistent with previous findings of no gender difference with respect to F0 in children younger than 12 years of age [15,31,32,33]. Some studies have found that F0 differences do not emerge before the age of 7 [34]. Nevertheless, since the age of the children in our study fell below age 7, no main effect of gender on F0 corroborated the results of earlier studies [15]. On the other hand, significant gender effects on vowel formants were found, which is also consistent with previous studies’ findings [10,31]. Gender differences in children’s formant frequencies may be attributable to vocal tract size, particularly the pharynx [10,31,35,36]. Our results are in line with those of previous studies reporting that adults can perceive the gender of children as young as 4 years of age. In Figure 2, the size of the vowel space is greater for girls than for boys. Thus, adults’ perception of gender from children’s speech sounds is strongly related to formant frequencies.

When it comes to differences based on vowel types, the results of our study diverge from those of previous studies. For example, Ref. [35] found that the gender effect appeared mostly in non-high and non-back vowels. Girls produce significantly higher F1 for the low-central vowel and higher F2 for the low-central and mid-front vowels. This result was consistent with [35,37], in which gender distinctions were reported to be small for high vowels, but large for lower vowels. Ref. [33] also found that the difference in the F1 value of [a] productions between Korean-speaking boys and girls was the greatest among vowels. However, this study showed that the distance of the high front vowel between boys and girls along the F2 dimension was as long as that of the low vowel along the F1 dimension (Figure 2). The different formant values between boys and girls could also indicate different articulatory gestures between the two genders. The degree of mouth opening might be a good index for displaying a gender-dependent articulatory gesture. Boys might tend to open their mouths more widely, resulting in a lower F1. In addition, boys might advance their tongues to a more anterior position than girls, especially for high front vowels. The vowel height and frontal quality are differentiated by around 24 months, and the roundness is completed after 36 months. Vowel development is also reported to precede consonant development [32]. Thus, the perceived vowel by adult listeners agrees with the production by children, as in Figure 3, except for [o].

The exceptional case of [o] can be explained if we take the ongoing sound change in Seoul Korean. Acoustic studies in the past decade documented a raised [o] by showing that the lowered first formants (F1) almost overlapped with those of the high back vowel [u] [6,24]. The spectral trajectories in [6] showed that the F1 and F2 between [o] and [u] were differentiated throughout the vowel midpoint although the trajectories gradually merged near the vowel midpoint in older male speakers’ productions. Based on the static spectral examinations in [6,24], we can conclude that low F1 values of [o] in our children’s speech confirm the vowel [o] rising, which reflects the ongoing vowel changes of their caregivers.

## 5. Conclusions

In this paper, we asked whether the acoustic vowel space of children’s speech would reveal gender differences and reflect the phonological patterning of adults’ speech. The results obtained in the present study is in line with the previous findings on American children such as [16] that F0 of Korean children as young as 6 years old does not serve as a reliable cue for gender distinction, but the formant patterning reflects the physiological difference between Korean girls and boys, functioning reliably as an acoustic cue for gender distinction. This finding for the role of formants in gender distinction differs from that in [14], in which no differences of vowel space in young Korean children (3- to 6-year-old) emerged due to gender. The present study also suggests that speech perception and production are indeed linked, but the link is not necessarily at the abstract level but rather at the physical level and constrained by phonetic vowel space and indexical information, such as gender.

The current study makes an important contribution to our understanding of the development and perception of children’s speech. Nevertheless, as developmental studies of Korean vowels are very limited, further investigation is necessary to confirm the findings presented herein. Specifically speaking, because we included two children in each age group and 36 vowel tokens (18 words × 2 children per age), it was not possible to conduct fine-grained acoustic analyses based on age group. Instead, we opted to collapse all data points for vowels and ran linear mixed effects models with vowel types and each individual child as random effects and gender as a fixed effect [29]. In further analyses, we will use an experimental design to assess developmental stages of children’s speech production and perception by including acoustic analyses of varied speech samples beyond the CVC word types across age groups. We will also conduct perception experiments, among others, with more tokens from children and listeners.

## Figures and Tables

**Figure 1 children-09-01690-f001:**
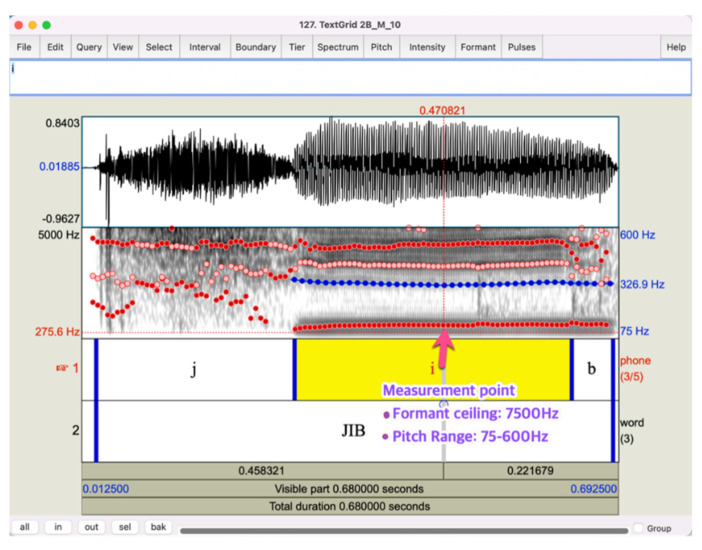
Screenshot of a sample speech file with annotation of a child’s production of a target word.

**Figure 2 children-09-01690-f002:**
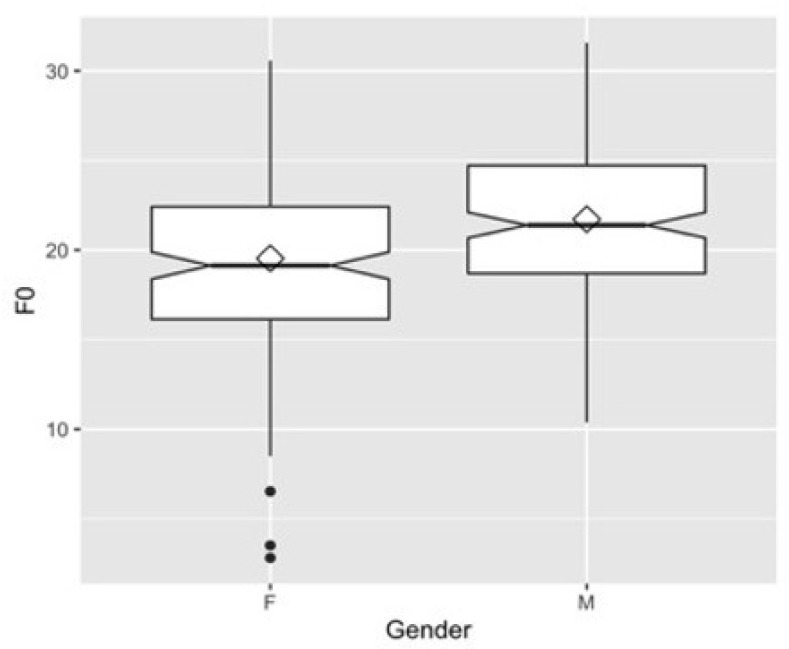
Box and whisker plot for F0 (in semitone) by gender.

**Figure 3 children-09-01690-f003:**
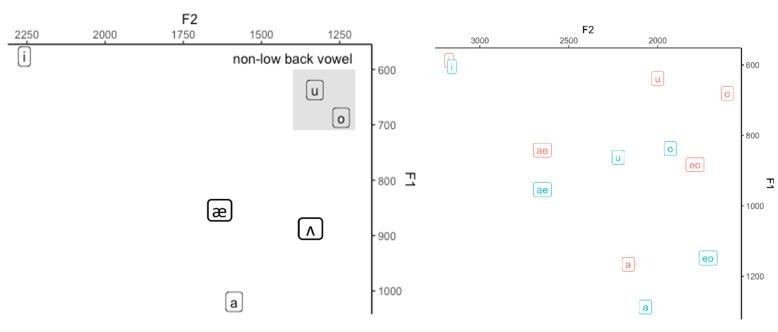
Vowel space for mean formants (**left**) and for mean formants of each gender (**right**). The blue vowels were produced by the girls and the red ones by the boys. [eo] in the right figure refers to [ʌ] in IPA.

**Figure 4 children-09-01690-f004:**
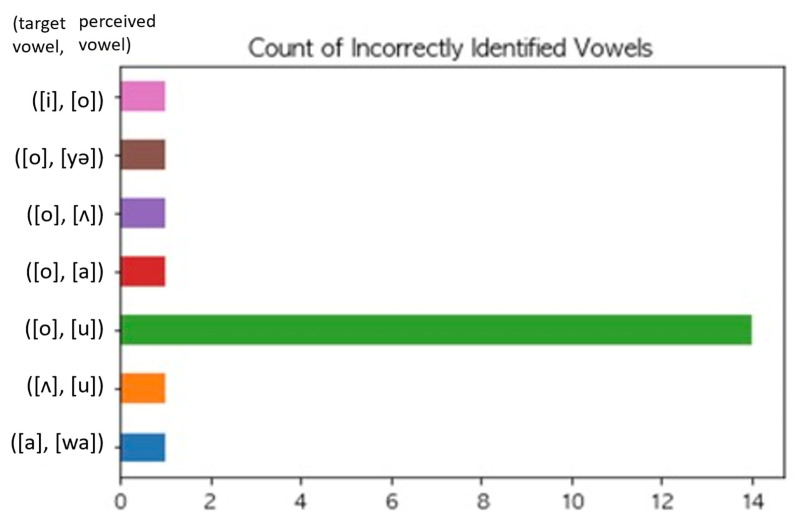
Count of incorrectly identified vowels.

**Table 1 children-09-01690-t001:** List of target words produced by children and perceived by adults.

Vowel	Sound	Meaning	Sound	Meaning
[i]	[kim] [ɹiŋ]	dried seaweed ring	[tɕip]	house
[æ]	[tɕ*æm]	Jam	[pæm]	snake
[a]	[p*aŋ] [hak]	Bread crane	[s*al]	rice
[ʌ]	[kʰʌp]	cup	[tʰʌk]	chin
[o]	[non] [k*otɕ]	rice paddy flower	[tɕʰoŋ]	gun
[u]	[son] [pʰul]	Hand glue	[t*oŋ] [mun]	Dung door

Tense and aspirated stops are marked with superscripted * and h symbols, respectively.

**Table 2 children-09-01690-t002:** Listener groups (A to E) and the age and gender of children in each group (M for male, and F for female).

Chilren’s Age	A	B	C	D	E
2	2F	2F	2M	2F	
3	3F	3M	3F		3F
4	4F	4M		4M	4F
5	5F		5M	5M	5M
6		6M	6F	6M	6F

**Table 3 children-09-01690-t003:** Fixed effects of gender on F0.

	Estimate	Std.Error	df	t Value	Pr(>|t|)
(Intercept)	19.563	1.01	20.0	19.2	2 × 10^−14^
GenderM	2.154	1.43	20.0	1.5	0.149

**Table 4 children-09-01690-t004:** Fixed effects of gender on F1.

	Estimate	Std.Error	df	t Value	Pr(>|t|)
(Intercept)	825.73	68.11	7.79	10.69	6 × 10^−6^
GenderM	−97.39	35.85	19.20	2.71	0.013 *

In the table statistical significance at the level of 0.5 is marked by *.

**Table 5 children-09-01690-t005:** Fixed effects of gender on F2.

	Estimate	Std.Error	df	t Value	Pr(>|t|)
(Intercept)	1645.57	145.09	7.01	10.2	1.8 × 10^−5^
GenderM	−159.2	65.27	18.88	2.4	0.024 *

In the table statistical significance at the level of 0.5 is marked by *.

## Data Availability

Not applicable.

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
