# Peer review of "Adults’ Perception of Children’s Vowel Production"

_children, 2022, doi:10.3390/children9111690_

Round 1

Reviewer 1 Report

Dear authors:

I have some comments related to the section "Materials and Methods". Could the authors provide more detail in relation to how the participants (20 children) were selected? Did the authors control for any differences among the children? For example, did any of them speak or were exposed to any additional language (other than Korean) at home or (pre)school? Were all of the children pre-schooled or schooled when they participated in the study?   

With regards to the data and the target words used in the study, the authors state that "the target words are familiar to children". (line 121). How did the authors account for this? How did the authors control for the fact that all target words were equally familiar to children? Did they use some sort of Corpus?  Providing this additional information for the section Materials and Methods would make it more sound and the readers would appreciate it. 

My other comment is related to the "Conclusions" section. The first paragraph seems a bit vague. Could the authors elaborate a bit more based on their findings and on previous research?  

Author Response

I have some comments related to the section "Materials and Methods". Could the authors provide more detail in relation to how the participants (20 children) were selected? Did the authors control for any differences among the children? For example, did any of them speak or were exposed to any additional language (other than Korean) at home or (pre)school? Were all of the children pre-schooled or schooled when they participated in the study?   

--> In the revised manuscript, we provided more detailed description of children and adult participants as you can see in section 2.1. 

With regards to the data and the target words used in the study, the authors state that "the target words are familiar to children". (line 121). How did the authors account for this? How did the authors control for the fact that all target words were equally familiar to children? Did they use some sort of Corpus?  Providing this additional information for the section Materials and Methods would make it more sound and the readers would appreciate it. 

--> In 2.2 in the revised manuscript, we provided a description on the familiarity of the target words. 

My other comment is related to the "Conclusions" section. The first paragraph seems a bit vague. Could the authors elaborate a bit more based on their findings and on previous research? 

--> We made the first sentence of the conclusions section clearer by providing a reference. 

Reviewer 2 Report

This is a highly interesting study and I appreciate the fact that it is done on an understudied language such as Korean. The study is well-grounded in literature. However, there are many points that the author(s) should address before publishing the paper. The main drawback is an insufficient description of the participants and the procedure.

 Aims

1. The study aims are expressed differently in different places. In the abstract the author(s) say(s) that one of the aims is to investigate "perception by inexperienced adults and also observed whether ongoing vowel changes in mid-back vowels affects adults’ perceptions when the vowels are produced by children." whereas at the end of Introduction they say that the study is aimed at investigating how "inexperienced listeners transcribe targeted vowels that children produce in monosyllabic words". Transcription may be one of the means to investigate perception but it is very tricky to use with linguistically naive participants. In the Material and Methods section we learn that the task for the adults was "to write down the monosyllabic words they understood." so this is not transcription as in phonology/phonetics study one would expect transcription to refer to phonological/phonetic transcription. The study aims need to be expressed more clearly and in the same way in the places where they need to be mentioned.

Materials and Methods

1. How was the children's familiarity with the words in the study checked?

2. It says that words were presented on screen for the children participants to produce. Probably it were not words as such but images with the objects that the words pertained to. This requires rewording. (Lines 125-126)

3. It says that when a child participant had problems producing the word, the research assistant asked the child to repeat after them. This is very problematic because of possible phonetic accommodation. Have/has the author(s) tried to compare the results for words repeated and words spontaneously produced?

4. In 2.3. Perception Study, it says that the stimuli were presented by means of a PowerPoint presentation and played through headphones. It is unclear what exactly was presented on the PowerPoint presentation. Was it an image or spelling or just a place to click for the sound?

5. A major drawback of this section is very scant information on the two groups of participants involved in the study. Both the children and adults' biodata need to described in detail.

Author Response

  1. The study aims are expressed differently in different places. In the abstract the author(s) say(s) that one of the aims is to investigate "perception by inexperienced adults and also observed whether ongoing vowel changes in mid-back vowels affects adults’ perceptions when the vowels are produced by children." whereas at the end of Introduction they say that the study is aimed at investigating how "inexperienced listeners transcribe targeted vowels that children produce in monosyllabic words". Transcription may be one of the means to investigate perception but it is very tricky to use with linguistically naive participants. In the Material and Methods section we learn that the task for the adults was "to write down the monosyllabic words they understood." so this is not transcription as in phonology/phonetics study one would expect transcription to refer to phonological/phonetic transcription. The study aims need to be expressed more clearly and in the same way in the places where they need to be mentioned.

--> It is true that orthographic transcription is not the same as phonological or phonetic transcription in English. But in Korean, the mismatch between written characters and pronunciation is minimal. And given the target words that we used in our study, we can safely regard the writing in Korean alphabet is regarded as a broad transcription in the phonological or phonetic transcription. 

  1. How was the children's familiarity with the words in the study checked?

--> Section 2.2 was revised so that we provide the information on the familiarity of the target words that we used in our study. 

2. It says that words were presented on screen for the children participants to produce. Probably it were not words as such but images with the objects that the words pertained to. This requires rewording. (Lines 125-126)

--> We wrote the sentences so that the procedure of data collection is more clear in section 2.2.

3. It says that when a child participant had problems producing the word, the research assistant asked the child to repeat after them. This is very problematic because of possible phonetic accommodation. Have/has the author(s) tried to compare the results for words repeated and words spontaneously produced?

--> We appreciate the comment. Given the difficulty conducting experiments with children as young as 2 or 3 years old, we haven't tried to compare the repeated words and spontaneous words. 

4. In 2.3. Perception Study, it says that the stimuli were presented by means of a PowerPoint presentation and played through headphones. It is unclear what exactly was presented on the PowerPoint presentation. Was it an image or spelling or just a place to click for the sound?

--> We provided more detailed description of the procedure in section 2.2

5. A major drawback of this section is very scant information on the two groups of participants involved in the study. Both the children and adults' biodata need to described in detail.

--> We elaborated on this biodata in section 2.1. and section 2.3 in the revised manuscript. 

Round 2

Reviewer 2 Report

The improvements based on my comments are acceptable.

Author Response

Thank you for your kind review. We make some further revisions based on your comments as well as the comments by the other reviewer, which can be found in the attached file. With all the best. 
